# Significance of Pulmonary Endothelial Injury and the Role of Cyclooxygenase-2 and Prostanoid Signaling

**DOI:** 10.3390/bioengineering10010117

**Published:** 2023-01-14

**Authors:** Rosa Nickl, Sandra Hauser, Jens Pietzsch, Torsten Richter

**Affiliations:** 1Department of Anesthesiology and Critical Care Medicine, University Hospital Carl Gustav Carus Dresden, Technische Universität Dresden, Fetscherstr. 74, 01307 Dresden, Germany; 2Department of Radiopharmaceutical and Chemical Biology, Institute of Radiopharmaceutical Cancer Research, Helmholtz-Zentrum Dresden-Rossendorf, Bautzner Landstrasse 400, 01328 Dresden, Germany; 3Faculty of Chemistry and Food Chemistry, Technische Universität Dresden, Mommsenstrasse 4, 01062 Dresden, Germany

**Keywords:** acute respiratory distress, cyclooxygenases, eicosanoids, endothelial barrier dysfunction, lung endothelium, lung infection, lung inflammation, prostanoid receptors, selective cyclooxygenase-2 inhibitors (COXIBs), traditional non-steroidal anti-inflammatory drugs (tNSAIDs)

## Abstract

The endothelium plays a key role in the dynamic balance of hemodynamic, humoral and inflammatory processes in the human body. Its central importance and the resulting therapeutic concepts are the subject of ongoing research efforts and form the basis for the treatment of numerous diseases. The pulmonary endothelium is an essential component for the gas exchange in humans. Pulmonary endothelial dysfunction has serious consequences for the oxygenation and the gas exchange in humans with the potential of consecutive multiple organ failure. Therefore, in this review, the dysfunction of the pulmonary endothel due to viral, bacterial, and fungal infections, ventilator-related injury, and aspiration is presented in a medical context. Selected aspects of the interaction of endothelial cells with primarily alveolar macrophages are reviewed in more detail. Elucidation of underlying causes and mechanisms of damage and repair may lead to new therapeutic approaches. Specific emphasis is placed on the processes leading to the induction of cyclooxygenase-2 and downstream prostanoid-based signaling pathways associated with this enzyme.

## 1. Introduction

The understanding and the importance of the endothelium has changed from a simple layer of cells lining the blood vessels to a significant complex network with countless fundamental functions. Its prominent involvement in the interplay of hemodynamic, humoral and especially inflammatory regulatory circuits has already led to a better understanding of the development and progression of various diseases, e.g., asthma, chronic obstructive pulmonary disease (COPD) [1] and acute respiratory distress syndrome [2]. Endothelial cells (EC) are in constant interaction with the surrounding blood stream, neighboring cells or the extracellular space, and fulfill both paracrine and endocrine functions. Therefore, the properties and appearance of endothelial cells are very heterogeneous. The layer thickness of the endothelium varies from 0.1 µm to 1 µm [3]. In this context, the heterogeneity of EC is due to a variety of requirements related to different tissues (such as lung, skin, or brain) and vessel types therein, such as arteries, veins, or capillaries. The diversity and especially the regional specialization of the cells is based on the transcriptional profile of the individual cell groups and depends on the tissue and the surrounding structures [4,5]. Non-fenestrated endothelial layers are mainly found in the macrovascular system, as well as in the capillary system of some organs, including the heart, lung, skin or brain. The primary function of this continuous cell layer type is to provide a barrier function. In contrast, there are continuously arranged fenestrated endothelial cells as in renal glomeruli or discontinuously arranged cell layers disrupted by pores, e.g., in liver, spleen or bone marrow. Thus, cell migration and selective permeability can be mediated. The high endothelial plasticity is a result of the interplay of a variety of phenotypic influences (e.g., inflammation, trauma, aging) on the one hand, and molecular features, such as the expression of a wide variety of mediators, on the other [6,7].

Two alveolar endothelial cell subtypes have been subsequently classified.

General capillary cells are bipotent progenitor cells, replenishing alveolar endothelium during maintenance and repair. They are positioned in contact with stromal cells and regulate vasomotor tone. They may give rise to so-called aerocytes, which are the second part of the alveolar endothelial. These large complex cells extend several alveoli and form multicellular tubes. They are closely fused with AT1 cells and form extremely thin regions of the gas exchange surface [8]. These highly specialized cells exist only in the lungs and are also a major contributor to leukocyte trafficking. The alveolar membrane is continuously exposed to a wide variety of airborne pathogens and toxins. This can have far-reaching consequences for the integrity of this delicate complex consisting of alveolar epithelial cells, intermediate basement cells, and capillary endothelial cells. Endothelial dysfunction and endothelial damage resulting from viral, bacterial, and fungal infections, mechanical stress, or aspiration are increasingly recognized as a major health problem. Understanding the underlying mechanisms of damage and repair is of great importance for the development of new compelling therapeutic approaches.

In particular focus in this review are the cyclooxygenase-2 and prostanoid-mediated signaling pathways. A greater understanding of the functional expression of this enzyme in the lung endothelium during the course of lung disease during therapy and in the healing process may allow important differential diagnostic and therapeutic conclusions. Experience in recent years with anti-inflammatory treatments targeting COX-2 through selective COX-2 inhibitors, also called COXIBs, has shown that sole and sustained inhibition of the enzyme is not always clinically indicated. It is often important to modulate the activity of COX-2 to the right extent and in the right time window, an aspect whose biochemical basis is also considered in this work.

In this review article, we summarize recent developments. A PubMed database search was performed in the 2nd quarter of 2022 using key words “acute lung injury”, “acute respiratory distress syndrome”, “aspiration”, “cyclooxygenase”, “endothelium”, “infection (bacterial, mycotic/fungal and viral)”, “lung” and “ventilator induced lung injury (VILI)” linked to each other by AND and to the phrases “cancer”, “extracorporal circulation”, “transplantation”, “cardiothoracic surgery” with NOT as Boolean function.

The abstracts of the identified studies were reviewed; all studies that met a priori exclusion criteria were excluded, and the full text was examined for those remaining.

## 2. Pulmonary Endothelial Dysfunction as a Result of Infections

Lower respiratory tract infections remain one of the leading cause of death worldwide, regardless of age [9]. Pulmonary infection is the primary cause in up to 51% of patients with acute lung injury (ALI) [10]. Research on infection-induced alveolar damage has largely focused on the mechanisms and pathophysiological processes related to the epithelial cells of the lung.

### 2.1. Bacterial Infections

Endotoxin or lipopolysaccharide (LPS), a proinflammatory mediator derived from the cell envelope of Gram-negative bacteria, compromises EC barrier function in vitro and in vivo primarily through the activation of toll-like receptor 4 (TLR-4) by activating neutrophils, macrophages and other cells producing proinflammatory mediators and free radicals [11,12,13,14]. A central role is played by the activation of the pulmonary nuclear factor kappaB (NF-κB), followed by the increased synthesis and release of cytokines, such as tumor necrosis factor alpha (TNF-α). This, in turn, activates neutrophils and increases their expression of the adhesion molecule E-selectin, while macrophages respond by synthesizing and releasing additional cytokines (Interleukine-1 (IL-1), IL-6, IL-8, IL-10) and TNF-α [15,16]. During LPS-induced ALI, prolonged sequestration time and arrest-like dynamic behavior of neutrophils have been shown to lead to neutrophil entrapment in capillaries and arterioles [17]. The neutrophil entrapment is followed by permeability changes of the lung endothelial cells and edema formation [18,19]. Furthermore, peeling of the endothelial glycocalyx from lung EC occurs [20]. This process of shedding leads to the release of syndecan-1 and presumably acts as an adaptation to inflammation by limiting damage by restructuring and loosening intercellular junctions [21]. LPS-induced disruption of the EC barrier is critically dependent upon changes in the cytoskeleton of lung EC, including the activation of the rat sarcoma homologue (Rho) signaling, tyrosine kinase(s) and protein kinase C (PKC) pathways [22,23,24]. This triggers contraction, and, thus, the subsequent opening of paracellular avenues of permeability [25,26,27]. Different histone deacetylase (HDAC) isoforms are also involved in the LPS-induced regulation of the cytoskeleton structures and, therefore, EC barrier disruption through deacetylation mechanisms [28,29]. In vitro investigation of specific HDAC inhibitor has already been demonstrated to mitigate LPS-induced impairment of human lung EC; thus, it may further come into focus as a future novel targeted therapy [30]. Early inhibition of heat shock protein 90 (Hsp90), and concomitant disruption of the Rho signaling appears to prevent LPS-induced hyperpermeability and may potentially provide another approach for further research efforts on targeted treatment [24].

Lipoteichoic acid (LTA) and peptidoglycans are components of the cell wall of Gram-positive bacteria. Their inflammation-inducing pathway leads via TLR-2. It occurs through the activation of the myeloid differentiation primary response 88 (MyD88) signaling pathway, immediately resulting in NF-κB activation and cytokine transcription [31,32]. In vitro experiments indicate increased permeability through a mechanism mediated by reactive oxygen and nitrogen species in lung endothelial cells after LTA exposure, which are released by alveolar macrophages and other cross-talking lung cells [33]. Furthermore, bacterial infection leads to inflammatory processes of the pulmonary endothelium not only by cellular components, but also by various toxins (Table 1).

### 2.2. Viral Infections

The impact of the SARS-CoV-2 pandemic brought viral respiratory diseases back into the focus of research efforts. The disintegration of profound vascular functions characterizes the similar clinical appearance of fulminant viral infections: edema, hemorrhage, thrombosis, and organ hypoperfusion. While the SARS-CoV-2, Dengue, and Ebola viruses affect various cell types, endothelial cells are the major target for Hanta, paramyxo, and Influenza A, H5N1 viruses [49,50,51,52,53]. Viral attachment is mediated by viral surface proteins that bind to specific glycan or protein cell receptors, resulting in cellular and host specificity [54]. Nonspecific attachment via interactions with scavenger receptors or various carbohydrates has also been reported [55]. In viral infections such as H1N1 [53], and presumably in SARS-CoV-2 [56], lung endothelial cells are responsible for regulating the immune response, which consists of the recruitment of the innate immune cells and the production of innate chemokines and cytokines. Depending on the type of virus, release of a wide variety of cytokine groups occurs. However, a common feature of all recent pandemic viruses is that they induce an excessive early cytokine response [57,58]. In response to cytokines, EC produces enzymes that trigger glycocalyx detachment, leading to disruption of endothelial barrier function. Dengue non-structural protein 1 (NS1) directly induces glycocalix shedding of lung endothelium through the upregulation of cathepsin L, endothelial sialidases, and heparanase through TLR-4, similar to bacterial infection via LPS [59]. However, in *Hanta* pulmonary syndrome, dramatic disruption of vascular barrier function is mediated by direct viral manipulation of inter-cellular junctions. Surface proteins increase vascular endothelial growth factor (VEGF)-dependent vascular permeability through the inactivation of β3-integrins. Induction of the RhoA signaling pathway via the Hantavirus N protein causes an additive VEGF-independent enhancement of permeability [60]. However, other cells of the immune defense system may play distinct roles in the interaction with endothelial cells in various viral infections. In a mouse model of experimental *human Metapneumovirus (hMPV),* macrophages are required for early entry and replication in the lung. Therefore, alveolar macrophages appear to function as regulatory cells that promote clinical disease, airway obstruction, and general lung pathology [61]. These results contrast with other studies of RSV infection, belonging to the same Paramyxoidae family as hPMV. The role of macrophages, being exposed to viral pathogens of RSV, is to contribute to the antiviral innate immune response [61,62].

Secondary bacterial infection in the course of pneumonia initially caused by viruses is associated with increased mortality and morbidity rates [63,64]. The viral-bacterial interplay causes a fatal enhanced pro-inflammatory innate immune response in pulmonary EC [65]. In Influenza pneumonia, but especially in COVID-19, bacterial co-infection dramatically increases the mortality rate up to 75.9% [64]. Further mechanisms of lung endothelial damage elicited by variant virus species are shown in Table 2.

### 2.3. Mycotic Infections

Another dreaded disease is pulmonary mycotic infection. Invasive mycoses are a severe hazard and a major cause of morbidity and mortality, especially in immunocompromised patients. Invasive aspergillosis (IPA) is one of the most serious opportunistic infections, with mortality rates ranging from 35% to over 60%, and plays a particularly significant role in neutropenic or bone marrow transplant patients [78,79]. However, invasive fungal infection can also occur in immunocompetent patients. The host inflammatory response and virulence here depend, inter alia, on the composition of the fungal cell wall [80,81,82,83]. For example, the chitin components of *Candida albicans* induce IL-1β synthesis through macrophages [83]. The angioinvasive behavior of fungal pathogens leads to tissue destruction and local vascular thrombosis, thus enabling the fungus to invade tissues and spread hematogenously to other organs [84]. Hyphae of *Aspergillus fumigatus* stimulate EC to express the leukocyte adhesion molecules E-selectin, tissue factor and vascular cell adhesion molecule 1 (VCAM-1) and secrete the proinflammatory cytokines TNF-α and IL-8 [85]. *Candida* additionally induces the expression of ICAM-1 [86]. *Aspergillus fumigatus* is able to stimulate endothelial cells to express tissue factor, thus triggering and aggravating thrombosis [87]. A unique feature of *Aspergillus* and *Zygromycetes* species is the ability to carry out luminal as well as abluminal angioinvasion [88,89]. In this regard, abluminal *Aspergillus* infection causes greater expression of these procoagulant and proinflammatory genes (except IL-8) while causing less endothelial cell damage [90]. *Candida* and *Aspergillus spp.* induce various cell-damaging reactions, which are referred to as PANoptosis in their entirety. Several caspases (1,3,7 and 8), as well as mixed lineage kinase domain such as pseudokinase (MLKL) phosphorylation, are involved in this Z-DNA-binding protein-dependent activation leading to pyroptosis, necroptosis and apoptosis [91]. In IPA, toxic products from activated neutrophils additionally exacerbate fungus-induced tissue necrosis in non-neutropenic hosts [85,92]. Table 3 shows further exemplary pathogens of various invasive fungi. Some fungal species do not interact directly with the lung endothelium but use a Trojan horse transport mechanism to enter the bloodstream. For example, *Cryptococcus neoformans* [93] and *Histoplasma capsulatum* [94] induce their own phagocytosis through mononuclear cells of the immune system, e.g., neutrophils and macrophages. They can proliferate therein and use these host cells as shuttles into the vasculature. *Cryptococcus neoformans* can spread further through a damaged lung epithelium.

## 3. Pulmonary Endothelial Dysfunction as a Result of Mechanical Stress and Aspiration

In addition to infectious causes, mechanical stress is often another trigger for pulmonary endothelial damage. The epitome of such a burden is the mechanical ventilation of patients especially in pulmonary disease. Mechanical stress of the lung parenchyma is a major factor determining ventilator-induced lung injury (VILI). Overdistension of the lung tissue results in the release of proinflammatory substances that further damage the lung and escalate the progression of the inflammatory response, contributing significantly to mortality [108]. Excessive alveolar stretch results in decreased barrier function of EC due to rearrangement of the cytoskeleton and progressive formation of paracellular gaps [109]. Increased expression of signaling and contractile proteins in pulmonary EC, including Rho GTPase, myosin light chain, myosin light chain kinase, zipper-interacting protein kinase, protease-activated receptor 1, caldesmon, and Hsp27 [110] and high-mobility group Box 1 (HMGB1) [111], leads to destabilization of cell–cell junctional cadherins, loss of peripheral organized actin fibers, and the development of central stress fibers due to mechanical stretch [112]. Disruption of endothelial-specific VE cadherin bonds is a central mechanism of altered pulmonary endothelial barrier function [113,114]. There is evidence that this mechanism enables transendothelial migration of leukocytes [115]. Presumably, this results in the local accumulation of leukocytes and platelets in microvessels. Loss of endothelial barrier function is a central pathophysiological mechanism in the development of ALI. The increased permeability of microvascular barriers leads to extravascular accumulation of protein-rich fluid in the interstitium and air spaces [116,117]. Another pathophysiological process of VILI is the activation and phosphorylation of NF-kB, triggering the activation of inflammatory cells and the formation of a chemoattractant gradient, inducing an inflammatory cascade [118,119]. The ongoing upregulation of proinflammatory cytokines in EC, including MCP-1, IL-6 and IL-8, promotes further local and systemic inflammation [120,121]. The mechanoreceptor TRPV4 (transient receptor potential cation channel subfamily V member 4) plays another key role in the development of VILI and, according to recent studies, induces pulmonary EC barrier disruption by disrupting mitochondrial bioenergetics [122]. VILI is opposed by macrophage TLR 4, which is involved in the recovery and resolution of VILI. TLR4 activated by Hsp70 conditionally promotes macrophage efferocytosis by suppressing the shedding of Mer receptor tyrosine kinase by inactivating ADAM17 signaling [123]. Another way to reduce ventilator-induced mechanical stress on pulmonary EC is through the concept of protective ventilation [124].

Hence, pulmonary endothelium is becoming the focus of scientific therapeutic approaches for the treatment of acute lung injury. One promising therapeutic option targeting the endothelium are ligands that bind to receptors on EC and activate stabilizing intracellular pathways, mediating cytoskeletal reorganization and tightening of the VE cadherin bonds. One approach to strengthen the endothelial barrier appears to be the lipid sphingosine-1-phosphate (S1P). This binds to specific endothelial receptors and induces cytoskeletal reorganization, activation of the Ras-related C3 botulinum toxin substrate (Rac), and promotes the formation of adherence junctions [125,126]. However, some S1P receptors can also trigger destabilizing activities depending on the time and duration of administration [127]. Another stabilizing agonist is the active fragment of the Robo4 ligand Slit (Slit2N), which inhibits tyrosine phosphorylation of VE cadherin [128]. Thus, the internalization of VE cadherin is suppressed and the loss of integrity of the endothelial barrier through proinflammatory cytokines is prevented [129]. Other stabilizing agonists, such as angiopoietin 1 and atrial natriuretic peptide, have been identified and require further investigation, particularly with regard to isolated endothelial effects [130].

Another significant mechanism of inflammatory damage to lung endothelium is aspiration that potentially causes chemical pneumonitis and pneumonia. Aspiration of gastric contents is considered an important risk factor for the development of ARDS. The hallmark of the inflammatory response to gastric acid is characterized by the acute infiltration of neutrophils in the alveolar space, alveolar hemorrhage, intraalveolar and interstitial edema [131]. Low pH acid aspiration primary injures the airway and alveolar epithelium but also results in endothelial injury [132]. Tumor necrosis factor α plays an essential role in this process. It leads to leukocyte activation and induces, among other things, the expression of endothelial adhesion molecules that lead to the migration of neutrophils into the alveoli [133]. The proinflammatory cytokine IL-8 is also essential in this process as a chemotactic for neutrophils and promotes migration through activated endothelium and activation of neutrophils [134]. It must not be neglected that gastric contents contain various other substances such as food residues, bacteria and their constituents, but also cytokines such as IL-1β. Therefore, all these factors tend to lead to complex inflammatory damage mechanisms in the pathogenesis of aspiration-induced lung injury [135,136]. Seawater, similar to the gastric content, is a hyperosmolar mixture. In addition to bacteria and viruses, it contains a high level of calcium and sodium and has a lower temperature [137]. Seawater aspiration primarily affects the regulation of pulmonary surfactant [138] and the alveolar epithelium directly [139]; furthermore, it leads to inflammatory reactions and destabilization of the endothelial barrier in the lungs. Endothelial semaphorin 7A promotes the inflammation and edema by increasing the endothelial permeability in seawater aspiration-induced ALI [140]. The expression of hypoxia-inducible factor 1 α (HIF-1α) is also upregulated due to seawater stimulus [140]. Another edema promoting factor is the increased expression of endothelial aquaporin 1 (AQP1), indicating elevated water permeability of the blood–air barrier [141]. 

The previous summary shows the multifactorial pathogenesis and signaling cascades that could be involved in inflammatory damaging processes of the pulmonary endothelium. In the next section, we will focus on the various COX signaling pathways that play a significant role in these processes.

## 4. Cyclooxygenase Signaling Pathways

### 4.1. Cyclooxygenases, Prostanoids, Prostanoid Receptors, and Downstream Signaling

Eponymous and central enzymes of the cyclooxygenase (COX) signaling pathways are the cyclooxygenases (COX), which are also known as prostaglandin G/H synthases. They catalyze the first two steps in the biosynthesis of prostanoids, well-known key players in the onset and resolution of inflammation, starting from arachidonic acid (AA), which is mobilized from the sn-2 position of membrane phospholipids via phospholipase A2 [142,143,144,145]. COX exists in three distinct isoforms: COX-1, COX-2, as well as a splice variant of COX-1, COX-3, which lacks enzymatic cyclooxygenase activity [146]. COX-1 is constitutively expressed in most tissues, compared to COX-2, which is regulated by transcription factors such as nuclear factor of activated T cells (NFAT), cAMP response element-binding protein (CREB), NF-κB, or hypoxia-inducible factors (HIF); therefore, it is induced by proinflammatory stimuli, cytokines, mitogens, activated platelets, as well as hypoxia and is constitutively expressed only in certain parts of the forebrain, cortex, hippocampus, and parts of the kidney [147]. Both COX-1 and COX-2 metabolize AA into prostaglandin G2 (PGG2) through their cyclooxygenase activity, followed by conversion of PGG2 into prostaglandin H2 (PGH2) via their hydroperoxidase activity. PGH2 then serves as a substrate for several downstream isomerases and synthases, whose expression depends on the cell type and is coupled to the COX isoform, in the generation of prostanoids: prostaglandins D2 (PGD2), E2 (PGE2) and F2α (PGF2α), thromboxane A2 (TXA2) and prostacyclin (PGI2) [143]. Dehydration also gives rise to the cyclopentane ring of PGE2 and PGD2 the cyclopentenone prostaglandins PGA2 and PGJ2 [148]. Prostanoids are synthesized in almost all mammalian tissues and regulate important homeostatic functions, such as the maintenance of the cardiovascular function, hemostasis or gastric epithelial cytoprotection [149,150]. In general, COX-2 exerts a dual role in the onset and resolution of inflammation, as evidenced by the fact that COX-2 expression peaks first at the onset of inflammation (within 2 to 6 h after stimulus) and second during the resolution phase (usually within 24 and 48 h). Moreover, both peaks coincide with the maximum PGE2 and PGD2 expression, respectively, indicating that COX-2 is proinflammatory during the early stage of inflammation, but acts in inflammation resolution at later stages [151]. Selected COX-2 signaling pathways are shown in Figure 1.

Prostanoids exert their function in an autocrine or paracrine manner by binding specific prostanoid receptors, which are classified into the five basic types DP, EP, FP, TP, and IP, for the PGD2, PGE2, PGF2α, TXA2, and PGI2 receptors, respectively, referring to their preferred binding partner. Prostanoid receptors belong to the class A, rhodopsin-like G protein-coupled receptors. Ligand binding induces both classical G protein-dependent pathways and G protein-independent pathways. Prostanoid receptors are promiscuous and often couple to more than one G protein; thereby, these trigger different signaling pathways. Further, G protein-independent signaling via a β-arrestin/Src complex, resulting in activation of the epidermal growth factor receptor (EGFR) and downstream Akt signaling, was shown for EP2 and EP4 receptors. However, prostanoid receptors function according to the principle of ligand-induced selective signaling; hence, downstream signaling is shaped by the prostanoid ligand. Therefore, another classification of prostanoids can be undertaken according to their influence on MAP kinase: MAP kinase inhibition through cAMP-dependent pathways (PGI2 and PGD2), MAP kinase activation (PGF2α and TXA2), and both, depending on the subtype of receptor bound (PGE2) [152,153,154,155,156]. The expression of all prostanoid receptors discussed above, except of DP2, has been shown in endothelial cells [157,158,159,160,161,162].

In addition to some prostanoids, low-dose oxidized phospholipids, such as oxidized 1-palmitoyl-2-arachidonoyl-sn-glycerol-3-phosphorylcholine (OxPAPC), can bind to EP4 receptor, which plays a central in promoting or disrupting the endothelial barrier function. In this regard, low-dose OxPAPC has enhancing characteristics on pulmonary endothelial barrier function, when sensed by the EP4 receptor, by tightening cell junctions and remodeling the cytoskeleton of EC [163]. Higher doses have the contrary effect, leading to disruption [164]. These opposite effects have also been experienced with PGE2, depending on the relative expression of the receptor subtype [165]. PGD2 was also found to have promoting properties on pulmonary endothelial barrier function in human pulmonary microvascular cells [166], whereas only moderate to no effects on lung endothelial cell integrity were demonstrated for human pulmonary arterial EC [167]. 

### 4.2. COX Pathways in Healthy and Injured Lung Endothelium

#### 4.2.1. COX Pathways in Healthy Lung Endothelium

Within the healthy blood vessel system, COX-1 is expressed in the endothelium where it couples with prostacyclin synthase to produce vascular and circulating antithrombotic PGI2, counteracting platelet COX-1-derived TXA2 [147,168,169]. In contrast, COX-2 is only slightly expressed constitutively in endothelial cells of the most blood vessels; however, these few constitutively expressed COX-2 molecules actually exert cardioprotective effects, as underlined by the severe cardiovascular side effects of selective COX-2 inhibitors (COXIBs) in long-term use [147]. Further, pulmonary endothelium-derived prostacyclin and prostaglandins contribute to pulmonary vasodilation at birth, thereby preventing the development of a persistent pulmonary hypertension of the newborn [170].

#### 4.2.2. COX Pathways in Injured Lung Endothelium

Many studies show overexpression of COX-2 and the resulting increased production of prostanoids during an inflammatory reaction in the lung, usually focusing on lung epithelial cells [171], fibroblasts, or inflammatory cells [172] (reviewed in [173,174,175,176]). For example, in a rat model of meconium aspiration, COX-2 expression has been induced in the respiratory epithelium and in alveolar macrophages [177]. In general, macrophages represent a major source of COX-2 and downstream signaling molecules in inflammatory conditions of the lung. While some studies have investigated prostanoids and downstream signaling, COX-2 expression in the lung endothelium in inflammatory situations is less known. 

##### Influence of Infections (Bacterial, Viral, Mycotic) on COX Pathways

LPS, the prototypical bacterial pathogen-associated membrane pattern (PAMP) and a major component of the outer cell membrane of Gram-negative bacteria, for example *Haemophilus influenzae, Pseudomonas aeruginosa, Bordetella pertussis*, or *Legionella pneumophila,* is known to induce the expression of COX-2 via an increase in phosphorylated p38 MAPK and a biphasic, glutathione-dependent activation of p42/44 MAPKs [178], as well as through the activation of NF-κB [179]. However, Gram-positive bacteria are also able to induce the upregulation of COX-2 and PGE2 in lung endothelial cells and alveolar macrophages via activation of MAPK, as shown by Szymanski et al. using a *Streptococcus pneumoniae* ex vivo infection model of human lung tissue [180]. *Streptococcus pneumoniae* further induces COX-2 in alveolar macrophages, thereby contributing to exacerbating fibrosis in a mouse model [181].

SARS-CoV-2 virus infections upregulate COX-2 expression, thereby, strengthening lung inflammation and injury observed in COVID-19 patients [182]. Further, hypoxia in severe COVID-19 may initiate the COX/thromboxane pathway in endothelial cells, leading to vasoconstriction and increased probability of thrombotic events [183]. Other viruses, such as RSV, HPIV-3 or H5N1, upregulate COX-2 in bronchial and bronchiolar epithelial cells and macrophages [184,185,186,187]. In a very recent paper, Gopalakrishnan showed that influenza infection induced COX-2 expression in inflammatory macrophages, which contributes significantly to influenza-induced lethality [188]. In contrast, poly(I:C), a viral PAMP that can be considered a synthetic analogue of double-stranded RNA present in, for example, reoviruses, which activates TLR-3, does not enhance COX-2 expression in mouse lung tissue. COX-2 knockout mice showed enhanced anti-poly(I:C) interferon responses, suggesting that COX-2 inhibitors might be a potential anti-viral therapy via boosting of the endogenous anti-viral response when provided soon after infection [189]. 

In mycotic infections with *Candida albicans* or *Candida auris,* the activation of MAPK pathways and proinflammatory signaling has been recognized, paralleled by the fungal evasion of the innate immune response [190]. A contribution to this evasion is probably provided by the COX-2/PGE2 axis and subsequent signaling via IL-33 that negatively regulates immune responsiveness, as shown in mouse models of fungal exposure with *Aspergillus fumigatus* [191] or *Alternaria alternata* [192]. This has been confirmed by in vivo COX-2 inhibition in these models, as well as in a mouse model of challenge with *Histoplasma capsulatum* [193].

##### COX-Pathways and ARDS

ARDS manifests in acute respiratory failure and increased-permeability pulmonary edema and is caused by direct injury to the lung, such as pneumonia or aspiration, as well as indirect mechanisms, such as sepsis or burn [194]. ARDS is usually accompanied by refractory hypoxemia and patients need mechanical ventilation [2]. Accordingly, the lung endothelium in ARDS exhibits inflammatory activation and loss of barrier function, in part mediated via COX pathways [195,196]. In this regard, it has been shown that COX-2 and PGE2 are significantly upregulated in remote ARDS after burn injury, which is induced by the binding of substance P (SP) to neurokinin-1 receptor (NK1R) and signaling via ERK1/2 and NF-κB [197]. PGE2, PGI2, and PGA2 do potentially exert barrier-protective and anti-inflammatory effects on lung EC in vitro and in vivo, tested in vitro in response to thrombin (barrier-disruptive), IL-6, and LPS (proinflammatory), and in vivo in a mouse model of acute lung injury (challenge with LPS 1mg/kg intratracheally) [167]. Ohmura et al. confirmed the barrier-protective effects of PGA2 in response to thrombin or LPS treatment in human pulmonary EC and in vivo and elucidated its mechanism of action via EP4 receptor and activation of Ras-related protein 1/Ras-related C3 botulinum toxin substrate 1 (Rap1/Rac1) GTPase and PKA targets at cell adhesions and cytoskeleton: VE cadherin, p120 catenin, zona occludens-1 protein (ZO-1), cortactin, and vasodilator-stimulated phosphoprotein (VASP) [198]. A different study showed that PGE2 binding to EP1 and EP4 receptor disrupts the barrier between pericytes and endothelial cells [199], underscoring the contrasting actions of PGE2 depending on EP receptor type as well as subsequent G protein coupling. PGD2 has also been shown to enhance the barrier function of human pulmonary EC challenged by thrombin, with these effects being mediated mainly by EP4 rather than the DP1 receptor. This highlights the fact that prostanoids and prostanoid receptors act in a pleiotropic manner [166]. In contrast, TXA2 induces disruption of the lung endothelial barrier function via TP receptor-mediated increase in intracellular Ca^2+^ concentration and activation of Rho kinase, resulting in actin cytoskeletal rearrangement, as investigated in vitro using HPAEC, HMVEC, and HUVEC [200]. On the other hand, 15d-PGJ2 is another anti-inflammatory prostanoid activating antioxidant genes via transcription factor nuclear factor erythroid 2-related factor 2 (Nrf2), and is generated by COX-2 in the late phase of inflammation as shown in a mouse model of acute lung injury induced by carrageenin [201].

##### COX-Pathways and Hypoxia

Hypoxia, as a consequence of, for example, aspiration, is known to induce hypoxic pulmonary vasopression (HPV), intending to match ventilation with perfusion and optimize gas exchange. If hypoxia is global, a generalized HPV occurs and may lead to pulmonary arterial remodeling and hypertension. According to animal studies of different species, in the acute phase of HPV, COX-2 is primarily linked to TXA2 synthase; therefore, it contributes to pro-proliferative pulmonary vasoconstriction, whereas in chronic HPV, COX-2 seemed to be coupled to PGI2 synthase, thereby promoting anti-proliferative pulmonary vasodilation (reviewed in [202]).

Another disease associated with hypoxia is the hepatopulmonary syndrome (HPS), which occurs in 15 to 30 % of patients with liver cirrhosis and is characterized by abnormal gas-exchange-induced hypoxia leading to intrapulmonary vasodilation, pulmonary vascular shunting and pulmonary angiogenesis. One major factor in the pathogenesis of HPS is assumed to be the accumulation of activated CD68+-macrophages in the lung microvasculature. Liu et al. found the COX-2/PGE2 signaling pathway to be activated in common bile duct ligation rat lung in vivo (an animal model of HPS) and PMVEC in vitro, which resulted in the accumulation of macrophages via an imbalance in the ratio of secreted bone morphogeneticprotein-2 (BMP-2) to crossveinless-2 (CV-2) [203]. The pathological pulmonary angiogenesis occurring in HPS, characterized by a collective directional migration of EC, is mediated by the activation of the PKC/Rac signaling pathway via the COX-2/PGE2 axis. This has been shown by Tang et al. in HPMVEC stimulated with HPS patient serum [204]. Figure 2 summarizes determinants of pulmonary endothelial barrier function with a relationship to COX and presents rsp. future therapeutic anti-inflammatory agents.

### 4.3. Therapeutic Approaches to Inhibit Cyclooxygenases in Lung Injury and Treatment Response

Unselective inhibition of cyclooxygenases through traditional non-steroidal anti-inflammatory drugs (tNSAIDs), such as, for example, aspirin, ibuprofen or diclofenac, is a long-established option to treat pain or inflammation. As tNSAIDs inhibit both the constitutive COX-1 and the inducible COX-2 they are, on long-term use, associated with gastric and renal side effects, mainly by disturbing the natural balance of prostanoids synthesized by COX-1. To overcome these problems, differently selective inhibitors of COX-2 (COXIBs) were designed based on structural differences between both COX isoforms: in its tertiary protein structure, COX-2 forms a side pocket, which COX-1 is lacking [205]. The majority of COXIBs, such as Celecoxib, Valdecoxib, and Rofecoxib, exhibit fewer gastric side effects compared to tNSAIDs, but increase the risk of cardiovascular side effects on long-term use [206]. This can be explained, amongst other concepts, by the disturbed balance of mainly COX-1-derived TXA2 and mainly COX-2-derived PGI2 towards thrombogenic TXA2, as well as increased metabolization of AA by 5-lipoxygenase into leukotrienes, established mediators of allergy and inflammation. Enhanced cardiovascular toxicity has even led to the withdrawal of rofecoxib and valdecoxib from the market [207,208]. Nevertheless, in the usage of tNSAIDs and COXIBs, the individual benefits and risks of the patients have to be considered. Especially in emergency and intensive care medicine, the advantages of COXIBs regarding their anti-inflammatory and analgesic effects often outweigh the risks. Celecoxib, etoricoxib, and parecoxib, a prodrug of Valdecoxib, are currently approved in Germany for the treatment of rheumatic diseases (e.g., such as arthritis and Bekhterev’s disease) and chronic pain, as well as for short-term postoperative pain management. In terms of inflammatory reactions in the lung, parecoxib is currently used in the treatment of ALI/ARDS and HPS.

With regard to potential further applications of COXIBs in lung inflammation, preclinical studies show beneficial effects of celecoxib or parecoxib administration. In a rat model of COPD, induced by nitrogen dioxide exposure, inhibition of COX-2 through celecoxib has been shown to reduce lung inflammation [209]. In a rat model of systemic inflammation induced by burn injuries, parecoxib applied intramuscularly after burn injury decreased plasma levels of inflammatory cytokines and lung myeloperoxidase level, and ameliorated systemic and lung inflammation [210]. Additionally, parecoxib could effectively reduce COX-2 expression, PGE2 level and lung injury induced by meconium aspiration in a rabbit model [211]. 

Patients receiving large volume and long-term mechanical ventilation due to, for example, lung injury or respiratory dysfunction are particularly susceptible to VILI, accompanied by local inflammation and lung injury up to systemic inflammation and multiple organ failure. Pharmacologic inhibition of COX-2 with CAY10404 significantly decreased COX activity and attenuated VILI in mice, attenuating barrier disruption and inflammation [212]. Meng et al. have confirmed this in a rat model of VILI, where parecoxib significantly improved gas exchange and survival by reducing edema, decreasing local and systemic inflammation and apoptosis [213]. Partly, this positive effect of COX-2 inhibition in VILI might be due to reduced recruitment of Ly6Chi monocytes that usually migrate to the site of lung injury and contribute to oxidative stress and inflammation [214]. Qin et al., recently showed in a rat model of lung ischemia-reperfusion injury (LIRI), which often occurs in pulmonary or cardiopulmonary surgeries and develops into ARDS, that a pretreatment with parecoxib ameliorates LIRI via reduction in oxidative stress and inflammatory mediators, and upregulation of heme oxygenase-1 in lung tissue [215]. 

In two different models of HPS, parecoxib was used to inhibit the activation of the COX-2/PGE2 signaling pathway, thereby, reducing the accumulation of activated CD68+-macrophages [203] as well as the directional migration of pulmonary EC [204], which are two major pathophysiological processes in the development of HPS.

Currently, cyclooxygenase inhibition through either tNSAIDs or COXIBs in order to reduce the cytokine storm and systemic inflammation in COVID-19 is a frequently investigated topic, and has to be discussed with regard to a potentially suppressed immune response due to COX-2 inhibition. Benefits for tNSAID use are likely, as they reduce the risk of development of severe disease in COVID-19 patients [216]. A recent study by Chen et al. showed that inhibition of COX-2/PGE2 signaling via ibuprofen and meloxicam, a moderately selective COX-2 inhibitor, had no effect on ACE2 expression, viral entry, or viral replication [182], thereby disproving the concerns that ibuprofen might lead to more severe infections via upregulation of ACE2. Based on a literature review, diclofenac (or other COX-2 inhibition medications) was considered recently for its anti-inflammatory and virus toxicity properties. For effectiveness and decreased risk, the use was recommended in high dose and in the early course of the disease (post-infection and/or symptom presentation) [217]. Recent evidence suggests to use of dual compounds COX-2 inhibitors/TP antagonist, providing anti-inflammatory effects and cardiovascular safety [218,219]. This was underlined by a clinical study indicating that adjuvant treatment with celebrex promotes the recovery of ordinary and severe COVID-19 [220].

## 5. Conclusions

Multiple factors, such as viral, bacterial and mycotic infections, aspiration or VILI, can severely damage the pulmonary endothelial tissue through different mechanisms, leading to consecutive hypoxemia and the development of ARDS. There is little diagnostic evidence to identify, for example, infection as a cause of pulmonary EC damage through inflammatory signaling pathways. Type and concentration ratios of the various prostanoid mediators involved in the inflammatory process or the extent of different up- and downstream signaling pathways cannot be determined analytically and a fortiori in the absence of clinical conditions, or only with great difficulty, to distinguish between tissue and organ-specific differences. However, it is precisely these differences that offer potential starting points for targeted individualized therapeutic intervention. An approach that serves to protect selected organs and tissues by modulating COX-2 activity and prostanoid signaling is indicated in addition to systemic anti-inflammatory therapy in many disease conditions, especially those described here. Such a targeted treatment is very difficult due to the complexity of the respective cellular interactions and specific microenvironmental conditions, especially in the presence of relevant comorbidities, and should be provided in the right time of the inflammatory process. Of course, the innate immune response and inflammatory processes are always an essential part of the healing process as well, and poorly balanced interventions can quickly lead to adverse effects. Therefore, each case must be handled individually and the type of drugs, possible concomitant drugs, dosage and duration of therapy has to be well-considered. In addition, there are approaches to investigate the anti-inflammatory effects of natural compounds. Components of pomegranate increased IL-10 gene expression and simultaneously decreased NO and TNF-α levels in LPS-exposed lung cells. However, the therapeutic potential of such compounds needs further investigation, specifically for acute lung endothelial injury [221]. Nevertheless, the development of new agents and the discovery and characterization of new drug targets will significantly advance this therapeutic approach. Promising approaches for selective inhibition of COX-2 include the use of hybrid or bifunctional compounds such as nitric oxide-releasing COXIBs. New treatment options should also be considered that allow for the local release of anti-inflammatory drugs via biomaterials, with the possibility of controlled optimal dosage and treatment duration as well.

## Figures and Tables

**Figure 1 bioengineering-10-00117-f001:**
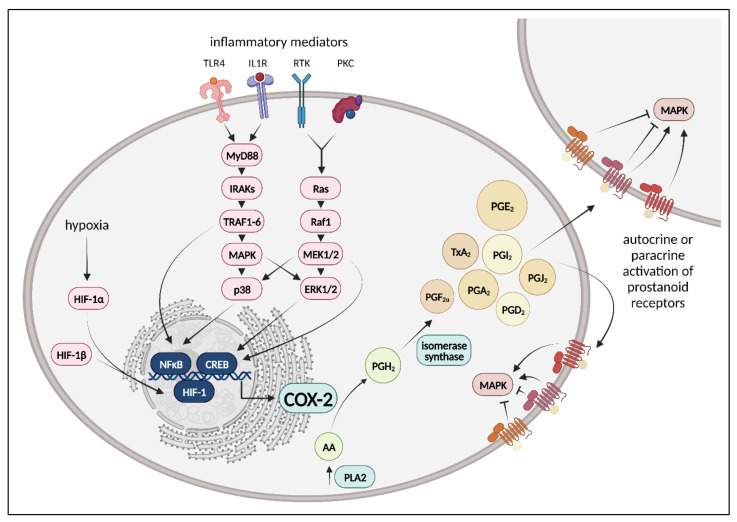
Schematic depiction of selected cyclooxygenase-2 (COX-2) signaling pathways. Ligand binding of inflammatory mediators to their receptors, such as pattern recognition receptors (Toll-like receptor 4, TLR-4), cytokine receptors (interleukine-1 receptor, IL1R), tyrosine kinase receptors (RTK) or protein kinase C (PKC) activates inflammatory signaling via myloid differentiation primary response gene 88 (MyD88)- or rat sarcoma (Ras)-dependent activation of several mitogen-activated protein kinases (MAPK), such as extracellular signal-related kinases 1/2 (ERK1/2) or p38. Thereby, activation of proinflammatory transcription factor nuclear factor κB (NF-κB) or cAMP response element-binding protein (CREB) is mediated. Further, hypoxia induces the activation of hypoxia-inducible factor-1α (HIF-1α) leading to the transcription of HIF-regulated genes. COX-2 is regulated by NF-κB, CREB, or HIF-1α, amongst others; therefore, it is expressed under proinflammatory or hypoxic conditions. COX-2 metabolizes arachidonic acid (AA), which is mobilized from the sn-2 position of membrane phospholipids through phospholipase A_2_ (PLA_2_) in a two-step reaction into prostaglandin H_2_ (PGH_2_), a substrate for several downstream isomerases and synthases in the generation of prostanoids: prostaglandins D_2_ (PGD_2_), E_2_ (PGE_2_), F_2α_ (PGF_2α_), A_2_ (PGA_2_), and J_2_ (PGJ_2_), thromboxane A_2_ (TXA_2_) and prostacyclin (PGI_2_). The size of the prostanoids shown corresponds to their relevance for inflammatory processes in the lung endothelium. Prostanoids act in an autocrine or paracrine manner by binding specific prostanoid receptors, which are class A, rhodopsin-like G protein-coupled receptors. Prostanoid receptors induce both classical G protein-dependent pathways and G protein-independent pathways, whereas downstream signaling is shaped by the prostanoid ligand and either activates or inhibits MAPK signaling. Mitogen-activated protein kinase kinase 1/2, MEK1/2; interleukine-1 receptor associated kinases, IRAKs; rapidly accelerated fibrosarcoma 1, Raf1; tumor necrosis factor receptor associated factor 1, TRAF1-6. Created with BioRender.com.

**Figure 2 bioengineering-10-00117-f002:**
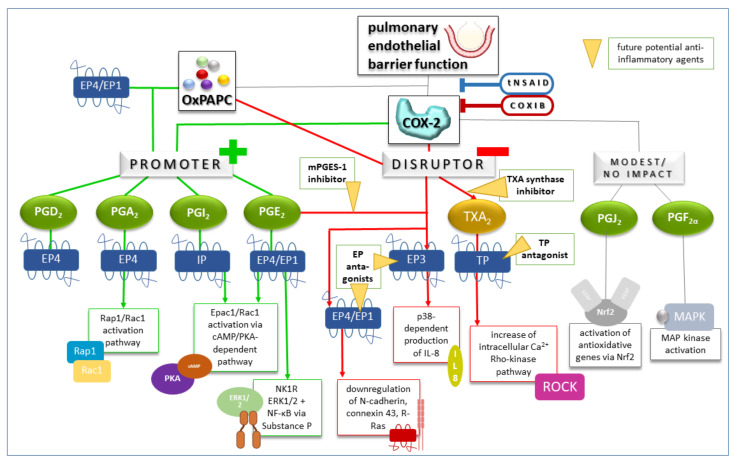
Determinants of pulmonary endothelial barrier function with a relationship to COX and present rsp. future therapeutic anti-inflammatory agents. Prostanoids as reaction products of the COX enzymes are presented as well as their subdivision into promoters and disruptors of the barrier function depending on the binding receptor and exemplary associated, with proven subsequent signaling pathways so far. Further, established and potential therapeutic anti-inflammatory agents and their targets are depicted. Cyclooxygenase, COX; selective inhibitors of COX-2, COXIB; exchange protein directly activated by cAMP 1, Epac1; prostaglandin E receptor 1, EP1; prostaglandin E receptor 3, EP3; prostaglandin E receptor 4, EP4; extracellular signal-regulated kinase 1/2, ERK1/2; interleukine 8, IL-8; prostacyclin receptor, IP; microsomal prostaglandin E 2 synthase-1, mPEGS-1; nuclear factor kappa B, NF-κB; neurokinin 1 receptor, NK1R; nuclear factor erythroid 2-related Factor 2, Nrf2; oxidized 1-palmitoyl-2-arachidonoyl-sn-glycerol-3-phosphorylcholine, OxPAPC; prostaglandin A2, PGA2; prostaglandin G2, PGD2; prostaglandin E2, PGE2; prostaglandin F2α, PGF2α; prostaglandin I2, PGI2; prostaglandin J2, PGJ2; protein kinase A, PKA; Ras-related protein 1, Rap1; Ras-related C3 botulinum toxin substrate 1, Rac1; traditional non-steroidal anti-inflammatory drugs, tNSAID; thromboxane receptor, TP; thromboxane A2, TXA2.

**Table 1 bioengineering-10-00117-t001:** Exemplary effects of various bacterial toxins on lung endothelial cells (EC).

	Toxin	Impact on EC	Sources
Gram positive			
*Staphylococcus aureus*	α-toxin	disruption of endothelial-cell tight junctions through (activating acid sphingomyelinase/release of ceramide) loss of barrier function through ADAM10 activation	[34,35]
*Streptococcus pneumoniae*	pneumolysin	activation of Ca^2+^-dependent enzymes, including PKC-α activation of the NF-κB and p38 MAP kinase pathways	[36,37,38]
*Listeria monocytogenes*	listeriolysin O	dysfunction in the ENaC channel	[39]
Gram negative			
*Pseudomonas aeruginosa*	exoenzyme S and T	activation of TLR-2 and -4 disruption of the actin cytoskeleton and interference of phagocytosis	[40,41]
	exoenzyme Y and U	microtubule breakdown and tau phosphorylation	[42,43]
	LasB	cleavage of VE cadherin	[44]
*Bordetella pertussis*	pertussis toxin	increase in PKC-mediated endothelial permeability	[45]
*Shiga toxin such as Escherichia coli*	subtilase cytotoxin AB	inhibition of protein synthesis and induction of vacuole formation	[46,47]
	shigatoxin 2	increase in cytokine and chemokine expression, e.g., TNF-α, IL-6, IL-8 inhibition of protein synthesis and induction of ribotoxic and ER stress responses	[46,48]

A disintegrin and metalloproteinase domain-containing protein, ADAM10; epithelial sodium channel, ENaC; elastase B, LasB; mitogen-activated protein, MAP; vascular endothelial cadherin, VE cadherin.

**Table 2 bioengineering-10-00117-t002:** Influence of selected virus species on pulmonary endothelial function.

	Primary Site of Lung Cell Damage	Specific Impact on Pulmonary EC	Sources
Orthomyxoviridae			
*Influenza A*	EC and epithelial cells	increase in cytoplasmatic translocation of High-Mobility Group Box 1 (HMGB1); release of HMGB1 via IL-6-receptor and activation of Janus kinase signal transducer and activator of transcription 3 (JAK/STAT3) signaling pathway activation of p38 MAPK and c-Jun N -terminal kinase pathways leading to cytoskeletal rearrangement and hyperpermeability via e.g., ERM (ezrin, radixin and moesin) phosphorylation	[66,67,68]
Paramyxoviridae			
*RSV*	EC and epithelial cells	upregulation of intercellular adhesion molecule 1 (ICAM-1)/vascular cell adhesion molecule 1 (VCAM-1) and E-selectin upregulation (dependent on protein kinase C (PKC), protein kinase A (PKA), p38 MAPK) promotes PMN transmigration	[69]
*Human Metapneumovirus*	epithelial cells, alveolar macrophages and dendritic cells	indirect impact on EC via triggering thymic stromal lymphopoietin (TSLP), IL-8 and IL-33 expression in epithelial cells, cytokines IL-4, IL-5, Interferon γ (IFN-γ), IL-10, and TNF-α	[61,70,71]
Coronaviridae			
*SARS-CoV2*	ciliated bronchial cells, alveolar cells and EC	dysfunction of bradykinin–kallikrein pathway and RAAS complex by angiotensin-converting enzyme 2 (ACE2) downregulation via ADAM17 mediated ACE2 shedding decrease in platelet-derived growth factor receptor β (PDGFR-β) and Angiopoietin I through pericyte loss	[72,73,74]
*MERS-CoV*	ciliated bronchial cells, alveolar cells and EC	upregulation of proinflammatory cytokines e.g., TNF-α, IL-6, CSF-1 and CSF-3, IL-32 endoplasmic reticulum stress and oxysterols enhance apoptosis	[75]
Bunyaviridae			
*Hantavirus species*	epithelial and EC	induction of transcriptional activation of VEGF and expression of B cell lymphoma 2 (Bcl2) gene activation of NF-κB induction of the expression of the chemokines RANTES (regulated upon activation, normal T cell expressed and presumably secreted) and enhancement of IP-10 infiltration of CD4^+^ and CD8^+^ T cells	[76,77]

**Table 3 bioengineering-10-00117-t003:** Effects of several fungal pathogens on lung endothelial cells in invasive mycoses.

Fungal Species	Pathogens	Specific Impact on Pulmonary EC	Sources
*Candida albicans*	mannan, chitins, β-1,3-glucans, β-1,6-glucans	recognition by pattern recognition receptors (PRR), e.g., mannose receptors and TLR-2 and -4, presumably enablement of adhesion to and transmigration across EC	[95,96]
	secreted aspartic proteases (Sap2, Sap6)	potent induction of IL-1β, TNF-α, and IL-6 production, e.g., through activation of NLRP3 inflammasome	[97,98]
	candidalysin	formation of pores in host cell membrane, induction of potassium efflux to cause NLRP3 inflammasome activation	[99,100]
	Als3 (agglutinin-like sequence 3)	Induces tyrosine phosphorylation of EC proteins, causing microfilament rearrangement resulting in pseudopod production and endocytosis	[101]
*Aspergillus fumigatus*	galactosamino-galactan	inhibition of translation by ribosome immobilization, induction of endoplasmic reticulum stress and triggers NLRP3 inflammasome activation	[102]
	dsDNA	induction of AIM2 inflammasome	[103]
	thaumatin-like protein CalA	interaction with integrin α5β1 on EC, inducing endocytosis	[104]
	gliotoxin	inhibition of NF-κB pathway, anti-angiogenetic activity	[105]
*Rhizopus oryzae*	coat protein homolog (CotH2 and CotH3)	binding via glucose-regulated protein 78 to EC and induction of endocytosis	[106,107]

Absence in melanoma protein 2, AIM2; nucleotide-binding domain and leucine-rich repeat family pyrin domain-containing 3, NLRP3.

## Data Availability

Not applicable.

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
