# Peer review of "Significance of Pulmonary Endothelial Injury and the Role of Cyclooxygenase-2 and Prostanoid Signaling"

_bioengineering, 2023, doi:10.3390/bioengineering10010117_

Round 1

Reviewer 1 Report

In this review, the authors describe the clinical significance of dysfunctions of the pulmonary endothel with a particular focus on cox2 induction.

The manuscript is well-organized and the topic is of sure interest to the readership of the "Bioengineering" Journal. Therefore, in my opinion, this review may be accepted for publication.

Author Response

Reviewer 1
English language and style
(x) English language and style are fine/minor spell check required

“Comments and Suggestions for Authors:
In this review, the authors describe the clinical significance of dysfunctions of the pulmonary endothel with a particular focus on cox2 induction.

The manuscript is well-organized and the topic is of sure interest to the readership of the "Bioengineering" Journal. Therefore, in my opinion, this review may be accepted for publication.”

Response:
Thank you for the comment on our manuscript. We have reviewed the manuscript again with regard to the English language and style and made appropriate changes marked in the manuscript. 

Reviewer 2 Report

Nickl et al review the role of COX2 and prostanoid signaling in pulmonary endothelial injury. This is an important and timely topic and the authors provide a manuscript that is well organized (e.g. section titles) and is supported by extensive literature searches on the topic. However, the scientific writing is quite poor; not in the English language context, but in that it often appears more as a listing of loosely related but disconnected facts/findings/observations related to the section title/topic. The writing is not flowing in a clear and logical manner, and it is often difficult to determine the main points the authors wish to convey.

Frequent writing deficiencies throughout the manuscript relate to:

1.       “Inaccurate or at best semi-accurate statements”:  For example, the first sentence of the Introduction implies that two decades ago (so in 2002), the endothelium was generally thought of as a simple layer of cells with singular/limited functions, which I think is incorrect.

2.       “Non-specific statements and unfinished thoughts”: For example, on page 1 the authors state in both the Abstract (Ln16) and the second sentence of the Introduction (Ln33) that endothelial dysfunction is associated with “numerous diseases”, but fail to elaborate further or provide any examples/context/references in either instance. As a result, these types of statements are of limited value to the reader.

3.       “Unclear reasoning and/or causalities”: For example, in the third sentence of the Introduction, the authors make the observation that “ECs are in constant contact with the bloodstream, neighboring cells and the extracellular space…”. It is unclear what point is being made here as, by default, all cells are in constant contact with the “extracellular space” (did you mean interstitial fluid?) and most mammalian cells are in constant contact with “neighboring cells”. Direct contact with the bloodstream is a unique property of ECs, but this fact is diluted here, and its significance for the role of ECs in lung disease is not further elaborated on.  The authors go on to state that because “ECs fulfill paracrine and endocrine functions (3rd sentence); therefore, the properties and appearance of endothelial cells must be heterogeneous (4th sentence)”. I would classify these statements as “true, true, but unrelated”. Yes, ECs can secrete hormones and mediators, and yes, ECs are heterogenous, but one is NOT necessarily related to the other. A single cell (or a clonal, homogenous population of ECs in culture) can exert multiple functions including barrier function, immune response, or hormone secretion. Seems to me that heterogeneity among ECs is predominantly driven by specific demands related to distinct tissues (e.g. brain versus lung) and vascular beds (e.g. arterial versus capillary versus venous; vessel diameter and branch points).

These types of deficiencies are numerous and continue throughout, and I soon gave up trying to list them all.  Thus, substantial editing (or better yet rewriting) of the manuscript is required to improve its value/impact for the reader.

Specific comments:
1. Pg1/2: As this manuscript is focused on lung injury, the discussion of ECs in other tissues (Ln 42-47) may not be necessary. On the other hand, the authors then switch to elaborate on two types of alveolar ECs, but miss any mention of the important distinctions between pulmonary arterial, capillary, and venous ECs.

2. Ln87/88: These first two sentences seem disconnected. What does “independently from age” signify?

3. Ln93: Please delineate what are the effects of LPS on ECs themselves (e.g. in in vitro cell culture) versus what effects are produced by LPS acting on other cell types.

4. Ln 97: NFkB is a transcription factor activated by LPS. It is not a cytokine (like TNFa) that is being “released”.

5. Ln100: what neutrophil entrapment? Context, causalities, and sequence of events are not clear.

6. Ln 105: “solid changes”?

7. Table 1 seems incomplete. For example, P.aeruginosa can express four exoenzymes (S, T, Y and U). Only S and T are listed, while the literature suggests that U and Y have the biggest impact in settings of PA lung infections.

8. Ln131: worldwide, widespread, and pandemic mean more or less the same thing.

9. Ln140: “ECs are the main culprits in regulating the immune response to viral infection”???? As a virus comes down the airway, the endothelial cells is neither the first cell/immune cell, nor the most important cell it encounters.

10. Ln 162: “Thereby, viral infections may not only resemble bacterial lung disease” Thereby? By what/How?

11. Ln 171: competitor?

12. Sections 2 and 3 may be shortened as there is a lot of discussion of literature related to cells other than ECs (e.g. macrophages) or the lung, or events not closely related to COX-related pathways (e.g. is there a significance of “endothelial prostanoid signaling” in settings of seawater aspiration).

  13. There are several loose ends in Fig. 1. For example, what ligand “binds to” and activates PKC. Not quite clear from the scheme which pathways activate which of the three transcription factors (NFkB, CREB, HIF-1).

14. Ln437: how does pericyte signaling in brain ECs relate to the lung endothelium?

15. Ln430: Compared to the idea to inhibit COX/COX2 as a therapeutic approach, approaches to treat lung injury patients with prostaglandin/prostacyclin therapeutics are severely underrepresented in this review.

16. Section 4.3.: It seems unclear to what extent the benefits of COX inhibitors are either derived from the inhibition of COX in ECs, or are due to changes of prostanoids acting on ECs (versus other cells).

Author Response

Reviewer 2
English language and style
(x) English language and style are fine/minor spell check required
Nickl et al review the role of COX2 and prostanoid signaling in pulmonary endothelial injury. This is an important and timely topic and the authors provide a manuscript that is well organized (e.g. section titles) and is supported by extensive literature searches on the topic. However, the scientific writing is quite poor; not in the English language context, but in that it often appears more as a listing of loosely related but disconnected facts/findings/observations related to the section title/topic. The writing is not flowing in a clear and logical manner, and it is often difficult to determine the main points the authors wish to convey.

Frequent writing deficiencies throughout the manuscript relate to:

1. “Inaccurate or at best semi-accurate statements”: For example, the first sentence of the Introduction implies that two decades ago (so in 2002), the endothelium was generally thought of as a simple layer of cells with
singular/limited functions, which I think is incorrect.

2. “Non-specific statements and unfinished thoughts”: For example, on page 1 the authors state in both the Abstract (Ln16) and the second sentence of the Introduction (Ln33) that endothelial dysfunction is associated with “numerous diseases”, but fail to elaborate further or provide any examples/context/references in either instance. As a result, these types of statements are of limited value to the reader.

3. “Unclear reasoning and/or causalities”: For example, in the third sentence of the Introduction, the authors make the observation that “ECs are in constant contact with the bloodstream, neighboring cells and the
extracellular space…”. It is unclear what point is being made here as, by default, all cells are in constant contact with the “extracellular space” (did you mean interstitial fluid?) and most mammalian cells are in constant contact with “neighboring cells”. Direct contact with the bloodstream is a unique property of ECs, but this fact is diluted here, and its significance for the role of ECs in lung disease is not further elaborated on. The authors go on to state that because “ECs fulfill paracrine and endocrine functions (3rd sentence); therefore, the
properties and appearance of endothelial cells must be heterogeneous (4th sentence)”. I would classify these statements as “true, true, but unrelated”. Yes, ECs can secrete hormones and mediators, and yes, ECs are heterogenous, but one is NOT necessarily related to the other. A single cell (or a clonal, homogenous population of ECs in culture) can exert multiple functions including barrier function, immune response, or hormone secretion. Seems to me that heterogeneity among ECs is predominantly driven by specific demands
related to distinct tissues (e.g. brain versus lung) and vascular beds (e.g. arterial versus capillary versus venous; vessel diameter and branch points).

These types of deficiencies are numerous and continue throughout, and I soon gave up trying to list them all. Thus, substantial editing (or better yet rewriting) of the manuscript is required to improve its value/impact for the reader.

Specific comments:

1. Pg1/2: As this manuscript is focused on lung injury, the discussion of ECs in other tissues (Ln 42-47) may not be necessary. On the other hand, the authors then switch to elaborate on two types of alveolar ECs, but miss any mention of the important distinctions between pulmonary arterial, capillary, and venous ECs.

2. Ln87/88: These first two sentences seem disconnected. What does “independently from age” signify?

3. Ln93: Please delineate what are the effects of LPS on ECs themselves (e.g. in in vitro cell culture) versus what effects are produced by LPS acting on other cell types.

4. Ln 97: NFkB is a transcription factor activated by LPS. It is not a cytokine (like TNFa) that is being “released”.

5. Ln100: what neutrophil entrapment? Context, causalities, and sequence of events are not clear.

6. Ln 105: “solid changes”?

7. Table 1 seems incomplete. For example, P.aeruginosa can express four exoenzymes (S, T, Y and U). Only S and T are listed, while the literature suggests that U and Y have the biggest impact in settings of PA lung infections.

8. Ln131: worldwide, widespread, and pandemic mean more or less the same thing.

9. Ln140: “ECs are the main culprits in regulating the immune response to viral infection”???? As a virus comes down the airway, the endothelial cells is neither the first cell/immune cell, nor the most important cell it encounters.

10. Ln 162: “Thereby, viral infections may not only resemble bacterial lung disease” Thereby? By what/How?

11. Ln 171: competitor?

12. Sections 2 and 3 may be shortened as there is a lot of discussion of literature related to cells other than ECs (e.g. macrophages) or the lung, or events not closely related to COX-related pathways (e.g. is there a
significance of “endothelial prostanoid signaling” in settings of seawater aspiration).

13. There are several loose ends in Fig. 1. For example, what ligand “binds to” and activates PKC. Not quite clear from the scheme which pathways activate which of the three transcription factors (NFkB, CREB, HIF-1).

14. Ln437: how does pericyte signaling in brain ECs relate to the lung endothelium?

15. Ln430: Compared to the idea to inhibit COX/COX2 as a therapeutic approach, approaches to treat lung injury patients with prostaglandin/prostacyclin therapeutics are severely underrepresented in this review.

16. Section 4.3.: It seems unclear to what extent the benefits of COX inhibitors are either derived from the inhibition of COX in ECs, or are due to changes of prostanoids acting on ECs (versus other cells).

Response:
We thank you for the very detailed and critical discussion of the chapter introduction. This should introduce the reader quickly and concisely to the topic, so to speak the tissue context, without going into too much detail
regarding mechanisms etc. Here we refer to other specific molecular biological review articles, e.g. DOI: 10.1038/nrm1357 or DOI: 10.1002/cphy.c180020.

We, as clinical scientists, have tried to present relevant things lucidly from our clinical-practical perspective.
Presumably, some presentations could be interpreted differently from a purely molecular biological point of view. In the following, we have attempted to refine some things in order to clarify the essentials.

1. We deleted the first part of the sentence in line 30: “The understanding and the importance of the
endothelium has changed from a simple layer of cells lining the blood vessels to a significant complex network with countless fundamental functions.”

2. We have added some more examples of the diseases in the introduction, including corresponding sources. Line 33: “Its prominent involvement in the interplay of hemodynamic, humoral and particularly inflammatory regulatory circuits has already led to a better understanding of the development and progression of various diseases, e.g. asthma, chronic obstructive pulmonary disease (COPD) [1] and acute respiratory distress syndrome (ARDS) [2].”. In the abstract, we would leave it like this for the sake of clarity, because otherwise it becomes too confusing for the reader.

3. In the introduction we wanted to give a general brief insight into the endothelial cells, their surrounding structures as well as their tasks, which were explained in the following sections specifically for the pulmonary endothelium and the diseases associated with it.
We have added the following sentence to line 41 for desired clarification:
“In this context, the heterogeneity of EC is due to a variety of requirements related to different tissues (such as lung, skin, or brain) and vessel types therein, such as arteries, veins, or capillaries.”

Specific comments:
1. In the introduction, we have deliberately refrained from going into further detail. Only at the beginning are individual organs listed to illustrate that the endothelium is ubiquitous in the human body. This review is primarily concerned with the capillary endothelium of the lung. Therefore, we
have briefly discussed the alveolar ECs in more detail. Topics such as pulmonary arterial hypertension were not considered further in this review and are therefore beyond the scope of this paper, including the introduction.

2. Mortality is the number of deaths in a given period per 1,000 individuals in a population. The period is usually defined as 1 year. We changed the sentence to: “Lower respiratory tract infections remain one of the leading cause of death worldwide, regardless of age”.

3. In vitro does not refer to living subjects. Therefore, the following section regularly refers to in vivo results and thus to clinically relevant results, which, as mentioned above, should be considered paramount for this review. For specific aspects in vivo and in vitro results on LPS-induced changes on endothelial cells, there are other reviews such as
https://journals.physiology.org/doi/full/10.1152/ajplung.00519.2016 ;
https://doi.org/10.1586/ers.10.71.

4. We corrected the sentence in the following way: Line 103 “A central role is played by the activation of the pulmonary nuclear factor kappaB (NF-κB), followed by the increased synthesis and release of cytokines, such as tumor necrosis factor alpha (TNF-α). This in turn activates neutrophils and increases their expression of the adhesion molecule E-selectin, while macrophages respond by synthesizing and releasing additional cytokines…“

5. We have supplemented the following sentence to be more precise and to clarify the aspect: “During LPS-induced ALI, prolonged sequestration time and arrest-like dynamic behavior of neutrophils have been shown to lead to neutrophil entrapment in capillaries and arterioles [18].“

6. We deleted the word “solid“ in Line 118.

7. We have added the exoenzymes u and y to the table. This is intended to show only examples of the specific effects of bacterial toxins on the endothelium.

8. We deleted the words “worldwide and widespread“ in Line 145.

9. We changed the expression to: “In viral infections such as H1N1 [54], but presumably also SARS-CoV-2 [57], lung endothelial cells are responsible for regulating the immune response, which consists of the recruitment of the innate immune cells and the production of innate chemokines and cytokines.“(Line 160)

10. We deleted the sentence.

11. We exchanged the word „competitor“ for „disease“ in the corresponding line 187.

12. The discussion in this review also involves interaction between endothelial cells to macrophages, as announced in the abstract.

13. PKC isoenzymes are activated by a plethora of molecules, including hormones, growth factors and neurotransmitters. When bound to their respective receptors, these molecules activate members of the phospholipase C family generating diacylglycerol. Diacylglycerol and, for some isoenzymes,
calcium ions are required for the activation of PKCs. We have added schematic depiction of diacylglycerol and calcium ions to the activated PKC, however, the detailed activation cascades are not shown in our Figure 1, as they are 1) textbook knowledge, 2) not focus of our review and 3) too rich in detail to show them here, which would entail to show activation cascades of the other receptors too.

14. In this study, brain pericytes + HUVEC were used. It is cited here, because it shows the influence of PGE2 and the EP receptor subtype on the interaction between these two cell types. The authors of the study discuss their findings in a general manner and not focused on brain tissue. They state that the disruption of pericytes mediated by PGE2 – EP signaling is a key process for vascular destabilization, which is one of the main focuses of this chapter.

15. Reviewed in DOI: 10.1126/sciadv.abp8322
https://doi.org/10.1177/088506660628704
We changed the title of the section to “Therapeutic approaches to inhibit cyclooxygenases in lung injury and treatment response”.

16. It derives from the combination of all mentioned issues.

Reviewer 3 Report

The manuscript is very well written. My evaluation clearly indicates that the scientific information provided by the authors is of high value and interesting enough for publication. However, there are several minor comments about the improvement of this investigation.

  1. The abstract focused mainly on the pulmonary endothelial injury, and the authors talk only in one sentence about COX-2. More information is needed to determine the relationship between the pulmonary endothelium and COX-2.
  2. Several keywords were used, and many of them are not mentioned in the abstract or title. Keywords are the most important words or terms in the title and abstract.
  1. Check if you can enrich your review with these citations:

https://doi.org/10.1016/j.biopha.2022.113667

https://doi.org/10.1016/j.biopha.2022.113721 

  1. Please provide the full form of abbreviations at the time of first use; thereafter, use the abbreviation. Check the article.
  2. Please review the English grammar and spelling in the article.

Author Response

Reviewer 3
"Comments and Suggestions for Authors:
The manuscript is very well written. My evaluation clearly indicates that the scientific information provided by the authors is of high value and interesting enough for publication. However, there are several minor comments about the improvement of this investigation.

1. The abstract focused mainly on the pulmonary endothelial injury, and the authors talk only in one sentence about COX-2. More information is needed to determine the relationship between the pulmonary endothelium and COX-2.

2. Several keywords were used, and many of them are not mentioned in the abstract or title. Keywords are the most important words or terms in the title and abstract.

3. Check if you can enrich your review with these citations:
https://doi.org/10.1016/j.biopha.2022.113667
https://doi.org/10.1016/j.biopha.2022.113721

4. Please provide the full form of abbreviations at the time of first use; thereafter, use the abbreviation. Check the article.

5. Please review the English grammar and spelling in the article.”

Response:
Thank you for this constructive comment.
We added the following statements:

1. We have included more precise details in the main manuscript and only wanted to provide a general outline here.

2. Keywords help indexers and search engines find relevant articles. If search engines can find the journal manuscript, readers can find it, too. Keywords should reflect the content of the manuscript by being specific to the field or subfield. They should just not appear in the title and abstract, but should provide information beyond that, since the title and abstract are already recorded bibliographically anyway. The keywords suggested here are in context and provide an extended 'search spectrum' for this purpose.

3. Thank you for pointing out the interesting articles. We have added, as important to cite in the context of our manuscript, https://doi.org/10.1016/j.biopha.2022.113721 in line 609 accordingly.

4. We have revised the manuscript with respect to this and inserted the abbreviations accordingly.

5. We have reviewed the manuscript again with regard to the English language & style and made appropriate changes.

Round 2

Reviewer 2 Report

accept in present form